# Molecular-Composition Analysis of Glass Chemical Composition Based on Time-Series and Clustering Methods

**DOI:** 10.3390/molecules28020853

**Published:** 2023-01-14

**Authors:** Ying Zou

**Affiliations:** School of Statistics and Applied Mathematics, Anhui University of Finance and Economics, Bengbu 233030, China; 20201131@aufe.edu.cn

**Keywords:** glass cultural relics weathering, analysis of time series, principal component analysis K-means

## Abstract

The weathering of ancient glass relics has long been a concerned. Therefore, a systematic and more comprehensive mathematical model with which to correctly judge the category of ancient glass products whose chemical composition changes due to weathering should be established. This paper systematically analyzes and studies the changes in the composition of ancient glass products as a result of weathering of. We first analyze the surface weathering of glass relics and its correlation with three properties and establish a multivariable time-series model to predict the chemical-composition content before weathering. Next, we use one-way analysis of variance for subclassification and, finally, we use a principal component analysis of the rationality, and change the significance level to determine its sensitivity, for the reasonable prediction of the chemical-composition content and classification to provide a theoretical basis for improving the model. This allows the model to provide reference values, which can be used in the protection of cultural relics, historical research, and other fields.

## 1. Introduction

Glass is precious material evidence of the trade between the Early Silk Road and the West. Cultural relics are the cultural heritage of a nation and the cultural carriers of civilization and the national spirit. Ancient glass is strongly subject to the influence of buried environments and weathering. Glass weathering is generally related to glass composition, glass surface chemicals, the environmental atmosphere of glass products, and other factors. During the process of weathering, there is a high degree of exchange between the internal elements and the environmental elements. In addition, the composition proportion changes. These changes affect the correct judgment of the category of weathered glass objects; the color and decoration on the surfaces of cultural relics surface cannot be used as the basis for judging their weathering, a large area of weathered cultural relics may still have unweathered areas. After the weathering of glass cultural relics, the differences between the chemical elements in different categories of cultural relics also have a certain relevance. With the development of society and the progress of science and technology, glass weathering, as a traditional topic, also urgently requires more rigorous and accurate methods of study. In this regard, the use of the quantitative analysis and type identification of the chemical composition is a good approach to the study of the weathering of glass cultural relics [1,2,3].

Gan Fuxi et al. combined X-ray fluorescence analysis, X-ray diffraction and laser Raman spectral analysis [4,5]. Li Qing-hui et al. studied the similarities and differences between flux, K_2_O-CaO-SiO_2_, PbO-BaO-SiO_2_ system glass, glass sand, and the Western Zhou, spring, and autumn [6,7]. Liu Song et al. discussed the application of a portable energy-dispersive X-ray fluorescence spectrometer to the analysis of the chemical composition of glass in ancient China [8,9,10].

Based on the related research on the chemical composition of glass, due to the disadvantages of the lack of an authoritative chemical-composition index, the research of domestic and foreign scholars is still limited. Most scholars analyze a small number of elements using qualitative methods or tools, with relatively little substantive research. This paper aims to organize the data of various chemical compositions and predict their content; unlike the work of other scholars, this paper focuses on the chemical composition of glass relics [11,12].

Through mathematical modeling, the classification and quantitative analysis of the known data on ancient glass relics, provide accurate results. Combined with the theoretical speed of chemical substances, this offers a theoretical basis for the reasonable prediction of chemical contents and the classification of cultural relics. In terms of the protection of cultural relics, we enrich the understanding of the preservation status quo and weathering mechanism, which provides a scientific basis for the protection-and-restoration scheme. Thus research expands the study of glass cultural relics and, at the same time, it has special significance in field operation, archaeological sites and cultural protection. (1) In this paper, various elements are incorporated into the same analytical framework and combined with the theoretical speed change of various chemical elements to systematically analyze the change trends of various elements, so as to supplement the domestic theoretical research on the changes in the chemical elements of glass relics and provide theoretical support for the research on the protection and restoration of glass relics. (2) From an empirical perspective, this paper uses the time-series model to predict the composition of various chemical elements, clusters the glass relics based on their characteristics, and provides relevant policy suggestions, so as to provide reference values for archaeological sites in China [13,14,15].

## 2. Data Sources and Basic Assumptions

The data are all from question C of the 2022 National College Students Mathematical Modeling Competition. To facilitate the research problem, the following assumptions are made [16]:(1)Environmental assumption: the glass is weathered under natural conditions, without special treatment, such as high temperatures;(2)Reasonability hypothesis: the changes between each element influence each other;(3)Unity assumption: overall unweathered, partial unweathered, overall weathered, local unweathered; overall weathering, local weathering, overall weathering, local severe weathering of the four stages of the same time;(4)Substantial assumptions: even at the beginning of the overall unweathered, local unweathered stage, weathering has begun;(5)Exclusivity assumption: all cultural relics are unaffected by their ornamentation, type and color, and the content of their chemical composition before weathering is a standard fixed value [17,18].

## 3. Data Pre-Processing and Visualization

### 3.1. Missing Data

Based on data, four values are missing from data “color” column: 19, 40, 48, 58. The common characteristic of four cultural relics was that surfaces were weathered. We speculated that the cause of cultural relics’ surface-weathering level is that their color cannot be observed. To support subsequent data processing, we used the four-color missing value as a reasonable supplement [19,20,21]

### 3.2. Data Preprocessing

According to the requirements, component proportions between 85% and 105% are regarded as effective data; the component proportions of the 15 and 17 sampling sites were 79.47% and 71.89%, respectively. Therefore, we deleted the data of the 15 and 17 sampling sites. Finally, the valid value obtained was 58.

### 3.3. Data Visualization

According to the existing data, the relevant information about the types, colors, patterns and weathering of different cultural relics was collected, and the types, colors and patterns were labeled as first-level characteristics, so that the data could be clearly displayed.

We used the chi-square test to analyze the correlation between the surface weathering of cultural relics and their characteristics. First, the hypothesis was true, and the value was calculated based on this premise, which represented the degree of deviation between the observed value and the theoretical value. According to the distribution and degree of freedom, the probability P of obtaining the current statistics and more extreme cases can be determined if the assumption holds. If the *p*-value is small, indicating that the deviation from the observed value is too large, the invalid hypothesis should be rejected to indicate significant differences between the comparative data; otherwise, the invalid hypothesis cannot be rejected, and the actual situation and the theoretical hypothesis represented by the sample cannot be considered.

In Table 1 and Table 2, it is possible to observe the relationship between the surface weathering of cultural relics and their characteristics. With a confidence interval of 95%, the *p*-value of decoration is 0.08, and the original hypothesis should be rejected; the *p*-value of type is 0.01, the original hypothesis should be rejected. If the *p* value of color is 0.42, the original hypothesis cannot be rejected; therefore, the weathering and glass types are significant, and the correlation with the decoration and color is not significant [22].

## 4. Predictions of Unweathered-Glass Chemistry Based on a Multivariate Time Series

### 4.1. Research Ideas

Based on the data presented in this paper, the surface weathering of the ancient glass cultural relics was divided into four stages: no surface weathering, local unweathered; surface weathering, local unweathered; surface weathering, local weathering; and surface weathering, severe local weathering. Secondly, each of the four stages was applied and the mean value was calculated to represent the criteria of each stage. On the premise of searching for relevant information as a theoretical support, combined with the theoretical rate of change of the chemical material, values to were assigned to each element. The composite score was calculated by multiplying the value coefficient of each element (the element-ratio weight of each glass type) by the mean and ranking it. According to the analysis, the lower the score, the greater the weathering degree. A total of 56 samples were processed in turn. The relationship analysis of the four-stage mean gave the initial value. This completed the multivariate time-series prediction. The specific process is shown in Figure 1.

### 4.2. Research Process

#### 4.2.1. Preparation of the Model

First, based on the characteristics of the data weathering (Appendix A), we divided the types of cultural relics into the following four stages: overall unweathering, local unweathering, overall weathering, local unweathering, overall weathering, local weathering, overall weathering, and local severe weathering.

Next, we performed a further analysis of the four stages. By referring to the literature, we determined that in glass products, after weathering, compounds composed of Si and Na elements can change significantly. For the group of elements comprising K, Ca, Al, and Pb, the content of the resultant compounds changes somewhat after weathering. We assigned the weights according to the proportion of changes in each element’s content after the weathering of glass product, according to the literature. Furthermore, according to the known literature, Si, Na, K, Ca, Pb, and Al are important variables in changes caused by glass weathering, and serve as weights according to the proportional changes before and after weathering in various types of chemical elements. In order to make full use of the form data, we weighted the remaining elements of the 14 chemical components. The final resulting weight ratios of the 14 chemical components are shown in Table 3 [23,24,25].

#### 4.2.2. Process of the Model

We organized the data and analyzed the data related to the time series with R, which can not only describe patterns in historical data over time, but can also be used for some studies and predictions. A multivariate autoregressive model was fitted using the VAR ( ) function in the multivariate autoregressive library “vars” in the R language. First, the image of each variable changed with the sampling point of the relic, as shown in Figure 2.

In the figure, it can be seen that the changes in SiO_2_ were relatively regular, the SnO_2_ and SO_2_ changes were relatively stable, while the K_2_O and Na_2_O changes were relatively stable in the early stage but fluctuated greatly in the later stage. The other elements constantly floated.

Second, we created a probability-density map, on which the abscissa range represents the value interval of each variable, the ordinate represents the probability of each variable taking the value, and the sum of the area between the curve and the x-axis is 1. In Figure 3, we can see where the values of each variable are mainly concentrated [26,27].

Next, we calculated the correlation between the variables, with the following formula:$$\rho_{xy}=\frac{\sum_{i=1}^{n}\left(x_{i}-u_{x})(y_{i}-u_{y}\right)}{\sqrt{\sum_{i=1}^{n}{\left(x_{i}-u_{x}\right)}^{2}}\sqrt{\sum_{i=1}^{n}{\left(y_{i}-u_{y}\right)}^{2}}}$$

According to the calculation results, when the absolute value is closer to 1, the correlation is stronger. When the result is closer to 1, the positive correlation is stronger, and when the result is closer to −1, the negative correlation is stronger. Based on the results, we created an analysis diagram of the correlation between the variables. In Figure 4, it can be seen that there were many obvious correlations between the variables. The result on the diagonal is the correlation of each element with itself; since the result was 1, we chose to only examine the diagonal results. We found that silica, potassium oxide, lead oxide, barium oxide, phosphorus pentoxide, and strontium oxide had an obvious correlation with the artificial sampling points. The artificial sampling points are the parts of glass relics that are analyzed according to the weathering and the content of each element [28].

Finally, we took various elements of each cultural relic as independent variables, predicted various elements in chronological order, and created the prediction plot. In Figure 5, as in the first figure, the black dots represent the true values, and the last five blue dots represent the predicted values. Based on the change results, we believe that the prediction results can better reflect the sequence-change characteristics [29,30,31].

#### 4.2.3. Solution of the Model

We created a five-step prediction of the time series, and the results are shown in Table 4.

The relationship-establishment analysis was used to obtain the regular launch initial value and to create the five-step prediction. For elements less than 0, we believe that a chemical reaction did not occur in the initial stage, which can be directly assumed to be 0. We compared the values that were not zero to the initial value derived based on the mean; we considered the values that were close to the initial value reasonable. The specific results are shown in Table 5.

## 5. Sub-Division of the Glass Types Based on the Clustering Algorithm

### 5.1. Research Ideas

➀ We pre-processed the data from Form 1 and preliminarily classified the categories of the cultural relics;

➁ We used a one-way analysis of variance to select the appropriate chemical composition for the cultural relics with high potassium and lead barium to obtain the first classification results;

➂ We used the K-means algorithm to subclass the glass types, and analyzes the specific division methods and results;

➃ The rationality of the results was supported if the form was classified again and the results were similar to the cluster junction; the sensitivity is judged by adjusting the significance level in the one-way analysis of variance.

### 5.2. Research Methods

#### 5.2.1. Selection of the Appropriate Chemical Composition

(1) Model Principle

One-way analysis of variance (ANOVA) refers to the method of analyzing the one-way test results and testing whether the factors have a significant impact on the test results.

Assuming that the collected data were derived from the sample values of S different populations (each level corresponds to one population), and counting the mean values of each population in one order, the following assumptions need to be tested:

Null hypothesis: $H_{0}:u_{1}=u_{2}=\cdots =u_{s}$.

Alternative hypothesis: $H_{1}:u_{1},u_{2},\cdots ,u_{s}.$ Not all are zero.

To reintroduce the horizontal effect, t $\delta_{j}\delta_{j}=u_{j}-u(j=1,2,\cdots ,s)$.
$$H_{0}:\delta_{1}=\delta_{2}=\cdots =\delta_{s}=0.$$

Thus, when true, the F-distribution-test statistic that needs to be followed by one-way ANOVA is:$$F=\frac{\left(S_{A}\right)/\left(s-1\right)}{\left(S_{E}\right)/\left(n-1\right)}=\frac{\frac{S_{A}}{\sigma^{2}}/\left(s-1\right)}{\frac{S_{E}}{\sigma^{2}}/\left(n-s\right)}\sim F\left(s-1,n-s\right).$$

Thus, with the significance level a, the rejection domain of the test problem is:$$F=\frac{\left(S_{A}\right)/\left(s-1\right)}{\left(S_{E}\right)/\left(n-1\right)}\leq F\left(s-1,n-s\right).$$

$F<F_{a.}$ At this point, the null hypothesis was rejected as showing significant differences between the samples.

(2) Model Building

First, we investigated whether the fourteen chemical components would have a significant effect on the glass-classification results of high-potassium types.

Therefore, we established the following assumptions:

Null hypothesis: The fourteen chemical components will not have a significant impact on the glass-classification results of high-potassium types.

Optional hypothesis: The fourteen chemical components will have a significant impact on the glass-classification results of high-potassium types [32,33,34,35,36].

In the ANOVA Table 6, SiO_2_, K_2_O, CaO, Al_2_O_3_, and Fe_2_O_3_ are less than 0.05. The null hypothesis is rejected in the belief that SiO_2_, K_2_O, CaO, Al_2_O_3_, Fe_2_O_3_, and high-potassium-type glass will affect the classification of high potassium type glass; that is, select these five suitable chemical components to subdivide the subclass of high-potassium-type glass.

Similarly, we investigated whether the fourteen chemical components would have a significant impact on the lead-barium-type-glass-classification results.

Therefore, we established the following assumptions:

Null hypothesis: The fourteen chemical components will not significantly affect the results of lead-barium-type-glass classification.

Optional hypothesis: The fourteen chemical components will have a significant impact on the results of lead-barium-type-glass classification.

In the ANOVA Table 6, it can be seem that SiO_2_, Na_2_O, Al_2_O_3_, CuO, PbO, BaO, BaO, P_2_O5, SrO, and SO_2_ are less than 0.05, and the null hypothesis that SiO_2_, Na2O, Al_2_O_3_, CuO, PbO, BaO, P_2_O_5_, SrO, SO_2_ and lead-barium-type-glass affect the classification of lead-barium-type glass can be rejected; that is, select the nine appropriate chemical components for lead-barium-type glass.

#### 5.2.2. Subclass Division

(1) Model Preparation

For the problem of using the above chemical components for each category, we use the k-means algorithm.

The K-value setting is the only defect of the algorithm. In order to improve the effectiveness of K value, we used the fast-clustering method to determine the value of K in K-means algorithm and obtained the K value of 3 through systematic clustering in SPSS software [37,38].

(2) Model Building

➀ We randomly selected K samples from the sample set as the initial mean vector;

➁ We calculated the distance of the sample from each mean vector and dividedthe sample into the phase according to the nearest mean vector from the sample cluster;

➂ After the classification, the central point of the category was redetermined and the mean of all samples in the category was made. For features corresponding to the new center point, the centroid of all samples in the class was applied;

➃ Steps 2 and 3 were repeated until the subclass subdivision of high-potassium glass with lead barium glass was completed.

(3) Model Solution

Based on the analyzed data, we solved the model using SPSS software and obtained the following results:

High potassium:

{18,7,27,10,12,9,22,21}, {16,14,3,1,4,5,13,6}.

Lead-barium:

{2,34,36,28,29,40,43,52,54,57}, {8,11,19,26,41,51,56,58,24,30},

{23,25,28,29,42,44,48,49,50,53,20,31,32,33,35,37,45,46,47,55}.

The detailed results are shown in Figure 6 and Figure 7.

### 5.3. Model Analysis

(1) Rationality Analysis

In order to verify the rationality of the classification results, we used the principal-component-analysis method to cluster the data and compare the observed classification results with the K-means classification results. If the comparison results were not very different, the classification results were reasonable; otherwise, the classification results were not reasonable. According to the results obtained from the above cluster analysis, the data were divided into two categories; therefore, we also extracted the same two categories using the principal-component-analysis method. The principal-component-analysis steps were as follows:

➀ With *n* cultural relics and *p* indicators, the initial sample matrix is:$$X=(X_{ij})n\times p,i=1,2,\cdots ,n;j=1,2,\cdots ,p$$

➁ Calculate eigenvalues of the inter-index correlation coefficient matrix R and eigenvector $e_{j}$ and obtain the principal component $W_{j}$: $W_{j}=Xe_{j}$.

➂ When the cumulative variance contribution of the j-th principal component is above 80%, take the first q principal components $W_{1},W_{2},\cdots ,W_{p}$; it is believed that the few q principal components reflect the information of the original *p* evaluation indicators. Formula for the cumulative-variance-contribution rate:$$a=\sum_{i=1}^{n}a_{j}$$

➃ The formula for studying the composite score is as follows:$$PC=aX_{1}+bX_{2}+\cdots +xX_{x}$$

Next, classifications were performed according to the coefficient size. Based on the coefficient of the principal components, we extracted the cultural-relic numbers with large coefficients and obtained the results presented in Table 7 [39,40].

Based on the above results, we found that although the results of K-means cluster analysis were slightly inconsistent, they were generally the same, which shows that the results essentially did not change with different methods, and that their rationality was strong.

(2) Sensitivity Analysis

During the extraction of the significant chemical-component content, the significance level of the one-way ANOVA was determined to be 0.05. To explore the sensitivity of our classification results, we adjusted the significance levels to 0.01 and 0.1, respectively, and the specific experimental procedures and results were as follows.

Significance level of 0.05: SiO_2_, K_2_O, CaO, Al_2_O_3_, and Fe_2_O_3_. Chemical composition in high-potassium-glass classification: SiO_2_, NaO, K_2_O, Al_2_O_3_, CuO, CaO, PbO, BaO, Fe_2_O_3_, P_2_O_5_, SrO, and SO_2_.

The ➀ significance level was 0.01.

For high-potassium glass, the significance level of the original chemical composition selected had less Fe_2_O_3_. For lead-barium glass, the extraction of chemical composition as less than the basis of the chemical composition of NaO, Al_2_O_3_, and SrO. Specific classification results are shown in Table 8.

The ➁ significance level was 0.1.

For high-potassium glass, the chemical composition extracted at this point was higher in SO_2_ than the previously extracted chemical. For lead-barium glass, the extracted chemical composition added CaO to the original significance level. The specific classification results are shown in Table 8.

The table shows that at the significance levels of 0.1 and 0.01, although the classification of the high-potassium and lead-barium glass was perturbed, fewer relics were disturbed; therefore, we believe that the sensitivity of the lead-barium-glass-classification law is low.

In conclusion, although changing the level of significance can affect the change in classification results, the number of changes is small and exerts a weaker impact on the population; therefore, we believe that the results obtained by the K-means classification are less sensitive.

## 6. Conclusions

By converting the relevant data to a time series, we can derive the initial value of each element variable. Based on the initial data of each element variable, combined with the existing element content, we can perform chemical reactions on the surfaces of glass artifacts. At the same time, based on the classification of glass with high potassium and lead barium, the use of subclass division can help us to obtain a more detailed understanding of glass artifacts. For a reasonable initial prediction of the chemical composition content, and to provide a theoretical basis for the division of cultural-relic categories, such studies ot only enrich the understanding of the preservation status and weathering mechanism of cultural relics but, at the same time, they are also of special significance in field operations, archaeological sites, and cultural-relic protection, among other applications. They also provides a scientific basis for the formulation of protection-and-restoration programs. Based on the research conclusion, the following policy suggestions are suggested:

First, increase the innovative research on glass relics. Most of the existing studies are on the protection and restoration of cultural relics. With the help of instruments and equipment, the research on specific elements should widen the research scope and conduct a more comprehensive study of glass relics.

Second, the richness of extenders combines color, category, decoration, and so on with chemical elements and classifies glass relics. This makes research on glass relics more relevant.

Third, we should attach importance to the coordinated development of the study of environmental systems and technological innovation. In the research on glass cultural relics, we should pay attention to environmental protection and encourage technological innovation through technical exchanges at home and abroad. This would highlight the development of glass technology at various points in time, as well as the integration of Chinese and Western glass technology.

## Figures and Tables

**Figure 1 molecules-28-00853-f001:**
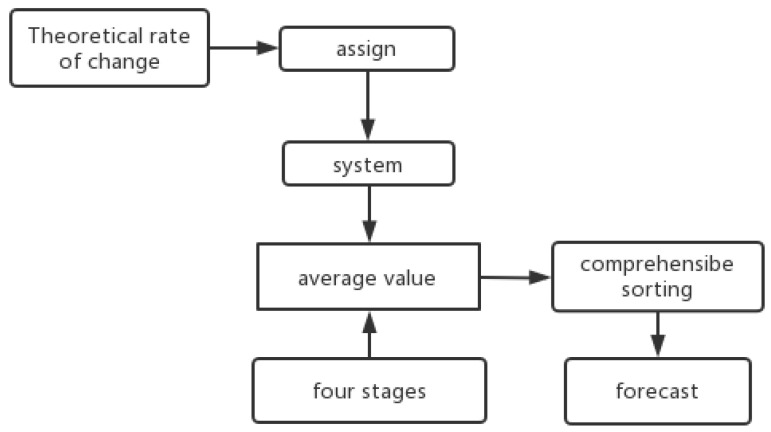
Process flow chart.

**Figure 2 molecules-28-00853-f002:**
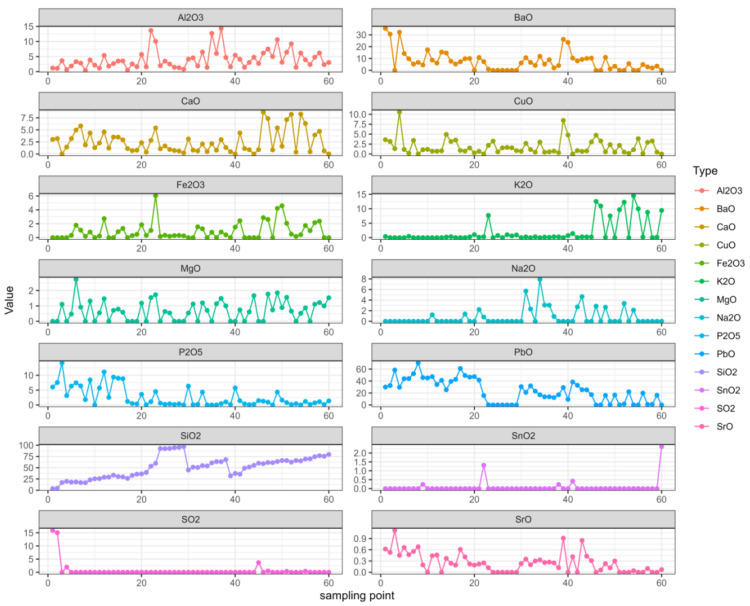
Changes in each variable.

**Figure 3 molecules-28-00853-f003:**
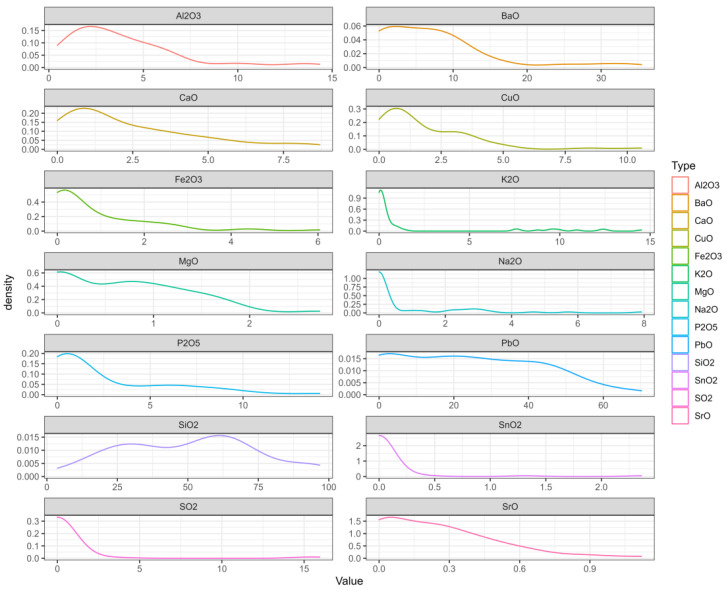
The probability-density map.

**Figure 4 molecules-28-00853-f004:**
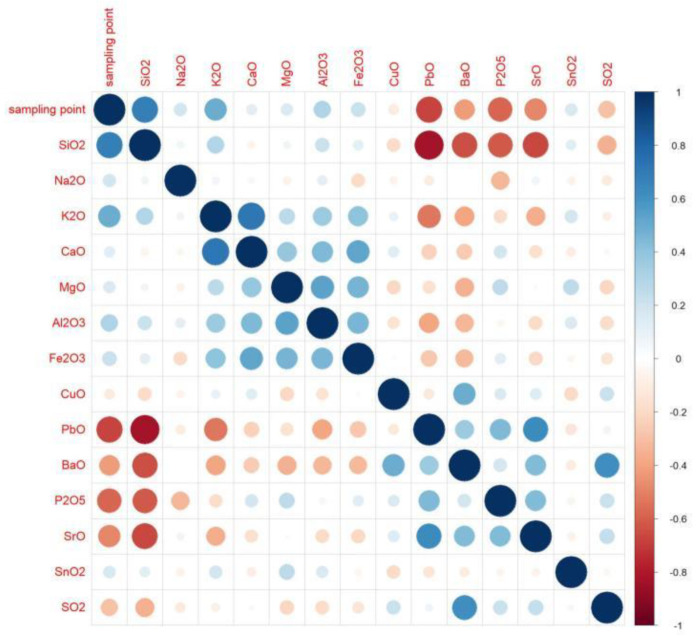
Correlation-analysis diagram.

**Figure 5 molecules-28-00853-f005:**
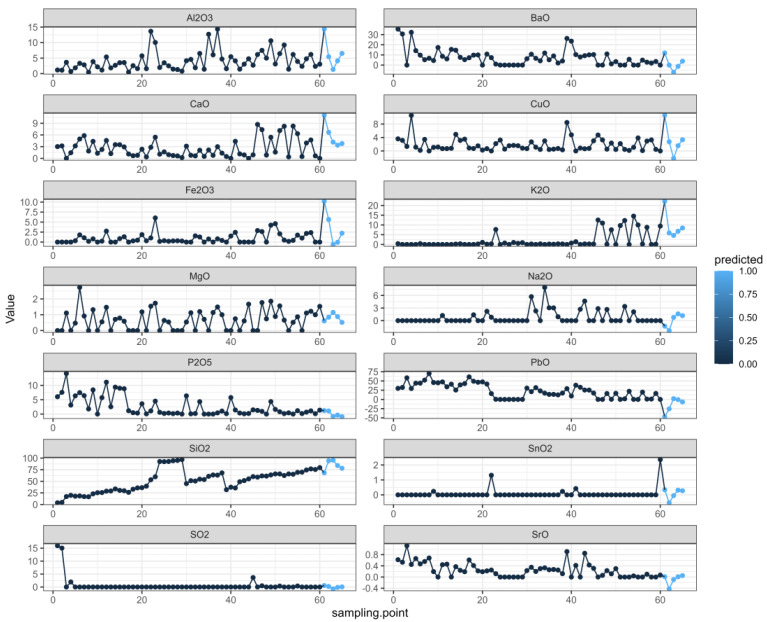
Prediction diagram.

**Figure 6 molecules-28-00853-f006:**
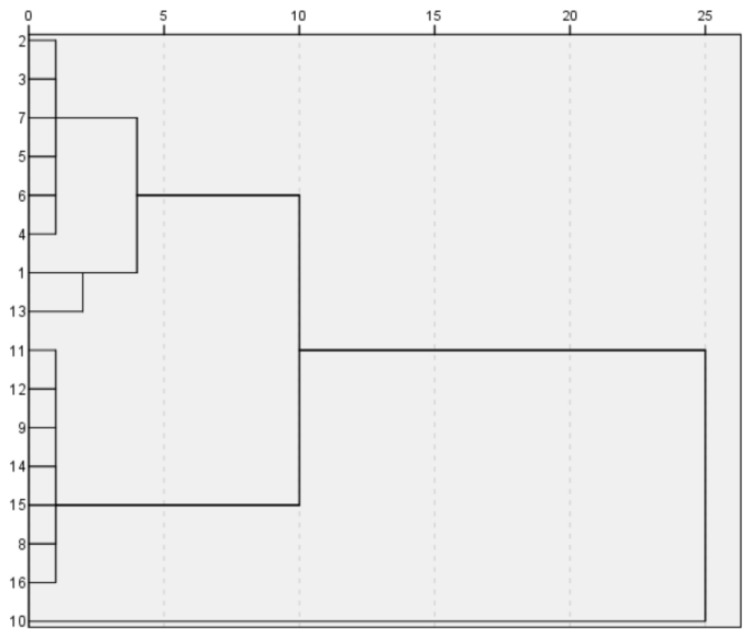
High-potassium-classification diagram.

**Figure 7 molecules-28-00853-f007:**
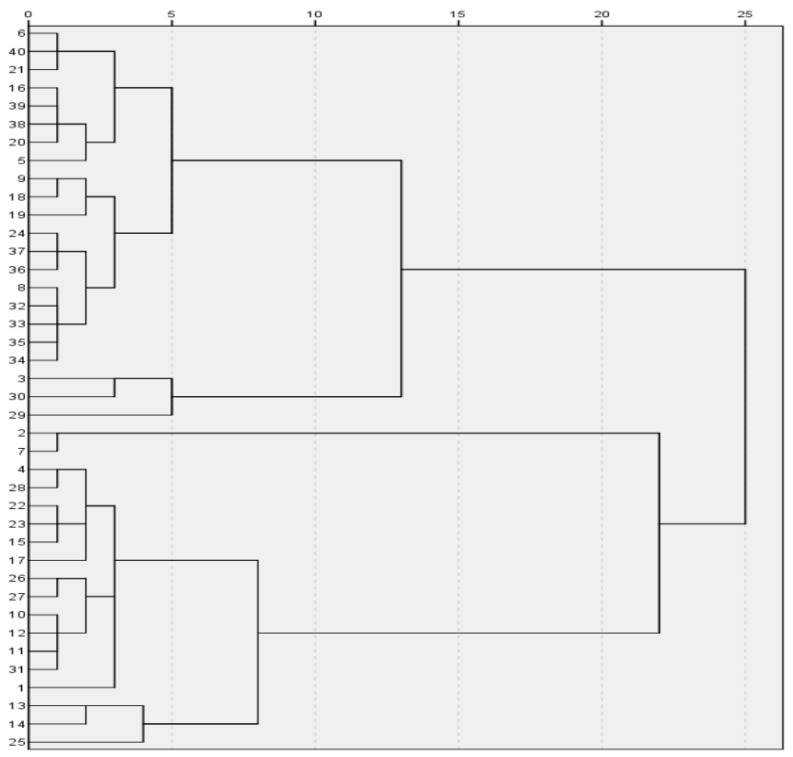
Lead-barium-classification diagram.

**Table 1 molecules-28-00853-t001:** List of cultural relics.

Characteristics of Cultural Relics		Surface Weathering	
First-Level Characteristics	Secondary Characteristics	Morals and Manners	Unweathered
ornamentation	high-manganese	6	12
	lead barium	28	12
type	A	11	11
	B	6	0
	C	17	13
pigment	black	2	0
	purple	2	2
	hispid arthraxon	0	1
	Blue–green	9	6
	pale green	1	2
	light blue	16	8
	deongaree	0	2
	dark green	4	3

**Table 2 molecules-28-00853-t002:** Chi-square test.

	Ornamentation	Type	Pigment
statistics	4.97	6.88	7.23
P price	0.08	0.01	0.42

**Table 3 molecules-28-00853-t003:** Weight ratios of 14 chemical components.

Element	Weight	Element	Weight
Si	0.358	Cu	0.016
Na	0.228	Ba	0.016
K	0.089	P	0.016
Ca	0.087	Sr	0.016
Pb	0.080	Sn	0.016
Al	0.030	S	0.016
Mg	0.016	Fe	0.016

**Table 4 molecules-28-00853-t004:** Prediction results.

Chemical Compound	1	2	3	4	5
SiO_2_	68.3	94.3	95.6	80.1	78.1
NaO	−1.3	−2.37	0.77	1.62	1.19
K_2_O	22.12	5.88	4.63	6.78	8.43
CaO	11.06	6.67	4.16	3.38	3.78
MgO	0.61	0.84	1.15	0.88	0.52
Al_2_O_3_	14.36	5.45	1.36	4.13	6.52
FeO	10.2	2.74	−2.2	1.56	3.32
CuO	10.63	2.74	−2.2	1.56	3.32
PbO	−46.7	−25.7	2.04	−0.63	−6.89
BaO	11.91	≤0.01	−7.06	−1.25	3.84
P_2_O_5_	1.21	1.08	−0.83	−0.36	−0.91
SrO	0.01	−0.42	−0.09	≤0.01	0.05
SnO2	0.34	−0.53	−0.05	0.31	0.27
SO_2_	0.56	0.21	−0.71	−0.12	0.05

**Table 5 molecules-28-00853-t005:** Results of each element component.

Element Component	SiO_2_	NaO	K_2_O	CaO	MgO	CuO	PbO
Results	83.26	0	9.67	5.81	0.8	3.21	0
element component	BaO	Al_2_O_3_	FeO	P_2_O_5_	SrO	SnO2	SO_2_
Results	1.49	6.36	3.12	0.04	0	0.07	0

**Table 6 molecules-28-00853-t006:** High-potassium-, lead- and barium-type-glass ANOVA table.

Chemical Composition	High Potassium		Lead Barium	
	F	P Price	F	P Price
SiO_2_	36.707	≤0.001	110.564	≤0.001
Na_2_O	3.973	0.166	7.454	0.035
K_2_O	35.837	≤0.001	0.086	0.870
CaO	42.173	≤0.001	7.781	0.081
MgO	2.230	0.158	0.541	0.365
Al_2_O_3_	13.587	0.002	7.839	0.026
Fe20_3_	6.851	0.020	0.062	0.364
CuO	2.252	0.156	2.617	≤0.001
PbO	0.833	0.377	73.763	≤0.001
BaO	0.048	0.830	2.816	≤0.001
P205	1.396	0.257	12.119	0.005
SrO	2.025	0.177	5.728	0.017
SnO_2_	1.000	0.334	0.400	0.781
SO_2_	4.114	0.062	1.593	≤0.001

**Table 7 molecules-28-00853-t007:** Main component analysis of high-potassium and lead-barium glass.

High Potassium						Lead Barium			
K-Means 1	PCA1	K-Means 2	PCA 2	K-Means 3	PCA 3	K-Means 1	PCA 1	K-Means 2	PCA 2
2	2	8	8	23	23	18	4	16	16
34	34	11	11	25	25	7	1	14	14
36	19	19	53	28	28	27	5	3	3
28	28	26	26	29	29	10	10	1	7
39	39	41	41	42	42	12	12	4	18
40	40	51	44	44	51	9	9	5	27
43	43	56	56	48	48	22	22	13	13
52	52	58	58	49	49	21	21	6	6
54	54	24	24	50	50	_	_	_	_
57	57	30	30	53	36	_	_	_	_
_	_	_	_	30	30	_	_	_	_
_	_	_	_	31	31	_	_	_	_
_	_	_	_	32	32	_	_	_	_
_	_	_	_	33	33	_	_	_	_
_	_	_	_	35	35	_	_	_	_
_	_	_	_	37	37	_	_	_	_
_	_	_	_	45	45	_	_	_	_
_	_	_	_	46	46	_	_	_	_
_	_	_	_	47	47	_	_	_	_
_	_	_	_	55	55	_	_	_	_

**Table 8 molecules-28-00853-t008:** Comparison of the various significance levels.

High Potassium				Lead Barium							
	0.05	0.1	0.01		0.05	0.1	0.01		0.05	0.1	0.01
01	2	2	2	02	1	1	1	39	1	1	1
03	2	2	2	08	2	2	2	40	1	1	1
04	2	1	2	11	2	2	2	41	2	2	2
05	2	2	2	19	2	2	1	42	2	2	2
06	2	2	2	20	2	1	2	43	1	1	1
07	1	1	1	23	2	2	2	44	2	2	2
09	1	1	2	24	2	2	2	45	2	1	2
10	1	1	1	25	2	2	2	46	2	2	2
12	1	1	1	26	2	1	2	47	2	2	1
13	2	2	2	28	2	2	2	48	2	2	2
14	2	2	2	29	2	2	1	49	2	2	2
15	1	1	1	30	2	2	2	50	2	2	2
16	2	2	2	31	2	2	2	51	2	2	2
17	1	1	1	32	2	2	2	52	1	1	1
18	1	2	1	33	2	2	2	53	2	2	2
21	1	2	1	34	1	1	1	54	1	1	1
22	1	1	1	35	2	2	2	55	2	2	2
27	1	1	1	36	1	1	1	56	2	2	2
_	_	_	_	37	2	2	2	57	1	2	1
_	_	_	_	38	1	1	1	58	2	2	2

## Data Availability

Data are contained within the article.

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
