# Peer review of "Molecular-Composition Analysis of Glass Chemical Composition Based on Time-Series and Clustering Methods"

_molecules, 2023, doi:10.3390/molecules28020853_

Round 1
Reviewer 1 Report
I have reviewed the manuscript which numbered molecules-2051512. The topic selection Angle of the manuscript is relatively unique, which is not only scientific and interesting, but also has good practical significance. Based on the perspective of Chinese glass objects, the chemical content of glass is predicted, and the cluster model is used to sub-classify the glass objects. While further classifying the glass relics, the author not only analyzed the stability, but also used the principal component analysis. The logical narrative of the manuscript is rigorous, the quality of the manuscript is good. The proposal is published after the minor revision. The specific modifications are as follows:
(1) The introduction part of the paper mentioned the literature review, but not, the literature review. Recommend writing down the comments in the introduction section.
(2) Clarify the practical significance and theoretical value of this paper in the introduction part.
(3) The research contribution proposed in this paper is not clear, and it is suggested that the research contribution should be refined and perfected.
(4) In the rationality test, the principal component analysis is used for the test, but only the test results are listed, please add the process of the principal component analysis.
(5) In the conclusion section, it is suggested to add some feasible policy suggestions.
Author Response
Response 1: Done. At the bottom of table 1, I have described them as primary features based on the existing data and data visualization to facilitate data display.(in red)
Response 2: Done. In the introduction, I add the practical significance and theoretical value of this study. (in red)
Response 3: Done. I refine the research contributions and add practical implications and theoretical value.(in red)
Response 4: Done.I add the procedure of principal component analysis. (in red)
Response 5: Done. In the last part of the paper, I make three feasible policy suggestions for the study of the chemical composition of glass relics. (in red)

Reviewer 2 Report
The work of Zou does not have a new theoretical development or improvement and the manuscript is not well written by the following points.
1. In table 1, th author has the column first level characteristics but it does not have any significant information because it is a classifier of the second level characteristic apparently “type” and “pigment” why it is assigned first level characteristics?
2. It is not supplied any Attachment 1, annex 1 and annex 2 in any supplementary material.
3. In table 2 by chi square test they regroup according to the first level characteristics and p price, where is defined before P price or explained in the text?
4. The number of 56 samples is a very low value to get deeply prediction and information then why are 60 samplings on the plots?
5. It is not explained how the author did the weight, since the var() function of R language did the job to get the multivariate autoregression, they should explain why the use R and which are the response and preddictors.
6. In fig 4 it is not explained details about the calculations of the correlation diagram and the physical meaning of 1 to -1 values, what is the meaning of sampling artifacts? The obvious diagonal of the correlation diagram.
7. The author order is confusing first it presents the prediction of fig 5 and later the author mention the ANOVA acronym without explanation in subsection 5.1 and in subsection 5.2 it explains the one way analysis of variance (acronym of ANOVA) and the clustering k-means method for subclass division. This order make the manuscript confusing.
8. The authors does not present a consisteng graphic to compare the real sampling relics versus the prediccted sampling of high potassium and lead barium.
9. Line 255 what means ANOVA one way ANOVA, lines 11 and 12 in the weathering of the weathering does not have any sense.
Author Response
Response 1: Done. At the bottom of table 1, I have described them as primary features based on the existing data and data visualization to facilitate data display.(in red)
Response 2: Done. I have provided Attachment 1, annex 1 and annex 2 in the supplementary material. (in red)
Response 3: Done. I added chi-square tests and P-values below Table 2 and added instructions.(in red)
Response 4: Done. I have already stated that in the article,According to the requirements, the component proportion and the data between 85% and 105% are regarded as effective data, and the component proportion of the 15 and 17 sampling sites is 79.47% and 71.89%, respectively. Therefore, we delete the data of the 15 and 17 sampling sites in Annex 2. Finally, the valid data obtained was 58. (in red)
Response 5: Done. I have added instructions to the setting of the weights, and we explain why R and the corresponding and predictor variables were used. (in red)
Response 6: Done. I provide formulas for the calculation of the correlation, as well as explanations for the correlation results-1 to 1. We then added instructions to the manual sampling points, as well as the correlation in fig 4. (in red)
Response 7: Done. After predicting the kinds of chemical elements of glass relics, we conducted variance analysis of each element. Part 5.1 is the step description, and part 5.2 is the initial step. More significant elements need to be selected for clustering according to the results of variance analysis, so the clustering results are more accurate. (in red)
Response 8: Done. I added the classification results of high potassium and lead barium in the text. (in red)
Response 9: Done. ANOVA one way ANOVA means one-way analysis of variance, I have made adjustments in the article.(in red)

Round 2
Reviewer 2 Report
The authors improved the manuscript according to my points but my main critic is focused in the analyzed data because it is not provided by scientific publications (absence of references) by analyzing the anexes and attachements files of the author, this work highly assumed that can be consistent with real data from publications but it is just assumptions.
The work is well corrected, however I recommend to publish this work for an educative journal and no published for a peer review JCR high impact factor journal as "Molecules".
I do not recommend to publish in this journal but I recommend to publish in "Education Sciences" of this editorial.